# Prevalence of Sleep Bruxism Reported by Parents/Caregivers in a Portuguese Pediatric Dentistry Service: A Retrospective Study

**DOI:** 10.3390/ijerph19137823

**Published:** 2022-06-25

**Authors:** André Brandão de Almeida, Rita Salgado Rodrigues, Carina Simão, Raquel Pinto de Araújo, Joana Figueiredo

**Affiliations:** Lisbon Pediatric Dentistry Service (SOL—Serviço Odontopediátrico de Lisboa), Santa Casa da Misericórdia de Lisboa, 1000-049 Lisbon, Portugal; rita.srodrigues@scml.pt (R.S.R.); carina.simao@scml.pt (C.S.); raquel.araujo@scml.pt (R.P.d.A.); joana.figueiredo@scml.pt (J.F.)

**Keywords:** sleep bruxism, children, pediatrics, clenching teeth, prevalence

## Abstract

The definition of sleep bruxism (SB) has changed over the years, and although it is no longer defined as a disorder, it is considered a risk factor that can result in prejudicial systemic and stomatognathic issues. The prevalence of SB in children is variable among studies, and its decrease during adolescence is a controversial matter among authors. We aimed to determine SB prevalence and assess its trend with age in a sample of pediatric patients who frequented the Lisbon Pediatric Dentistry Service. We conducted a retrospective observational study based on information provided by patients’ parents/caregivers and without examination or polysomnography examination. Data were collected between August 2019 and December 2020 by five dentists. A total of 1900 patients were included, 50.6% and 49.4% of which were male and female, respectively. Of the total sample, 334 (17.6%, 95% confidence interval of 15.9–19.4%) had bruxism, corresponding to 18.9% and 16.2% of male and female patients, respectively (*p* = 0.121). SB was reported in 20.7% of patients 0–6 years old, in 19.4% of those 7–11 years old, and in 14.6% of those 12–17 years old. In conclusion, SB showed a slightly higher prevalence among male pediatric patients, but further studies are needed to rule out confounding factors.

## 1. Introduction

Sleep bruxism (SB) is defined as “a masticatory muscle activity during sleep that is characterized as rhythmic (phasic) or non-rhythmic (tonic) movement”. In contrast, awake bruxism is defined as “a masticatory muscle activity during wakefulness that is characterized by repetitive or sustained tooth contact and/or by bracing or thrusting of the mandible”. Both exclude a movement disorder or a sleep disorder in otherwise healthy individuals [1].

SB is not considered a disorder but a risk factor in healthy individuals for a huge number of oral consequences, such as a loss of tooth structure and painful temporomandibular joints (TMJ) and muscles, which can result in a prosthodontic approach [2]. Contrary to the habit of grinding/clenching teeth during some moments of stress that can be stopped or consciousness of the habit can be gained, SB is a behavior totally independent of the conscious control of the child/person being controlled by the central nervous system [3].

The causes (or etiology) of SB remain unknown. Risk factors for SB may include emotional and psychological problems, such as anxiety and stress; personality factors, such as aggressivity; behavioral problems and hyperactivity; and poor sleep, nightmares, and snoring [2,3,4,5]. Concerning sleep quality, pediatric and adolescent bruxers showed a huge incidence of micro-arousals when a polysomnography was performed, which may happen at any stage of the sleep, occurring more frequently in stages 1 and 2 of non-rapid eye movement (REM) sleep [2,3,4,6]. Moreover, patients with attention deficit hyperactivity disorder (ADHD) treated with medication had a higher probability of developing bruxism when compared with pharmacologically untreated ADHD patients [4,7,8].

The diagnosis of SB is usually based on parents/caregivers’ reports and clinical examination. These can be complemented with polysomnography, which is the gold standard in SB diagnosis. However, polysomnography is expensive and has some limitations, namely, the fact that the children have to sleep in a laboratory [2,4,9].

The history (anamnesis) of the child, provided by their parents/caregivers, should include the sounds heard when the child is sleeping and their facial expressions in instances of clenching. The pain or discomfort is also important to record, as well as the frequency of headaches and the hypersensitivity to hot or cold food. The presence of parasomnias should also be recorded since an association between SB and sleep talking, in addition to other sleep disorders, has been found [4,10,11].

In clinical examination, it is important to detect SB-associated findings, such as tooth wear, gingival recession, fractured teeth or restorations, pain in the masticatory muscles and TMJ, TMJ sounds, and hypertrophy of the muscular system. Different diagnostic criteria have been suggested by several authors. Tooth wear remains a point of disagreement since it may be observed in disorders other than SB [2,4,6,12].

Due to the complexity of the etiology and the lack of total understanding of SB in children, there is no evidence of which approach is the most effective at controlling this parafunctional activity [13]. Different treatment approaches are available depending on the patient’s needs, including pharmacological therapy, surgical interventions, orthodontic intervention, physiotherapy, and the use of a bite plate [14,15].

The prevalence of SB varies from study to study, affecting 5.9% to 49.6% of children [4]. Additionally, while some studies have reported that the male sex is the most affected among children, this conclusion is not well established. A decrease in SB when reaching adolescence seems to be a tendency, but it is not also properly studied. Furthermore, while some studies show that the prevalence is higher in deciduous dentition, others reported a higher prevalence in the first phase of mixed dentition [7,16]. The differences in the numbers of prevalence may be explained by several factors, such as the different criteria used, with the highest rates obtained when using information that only relied on the caregivers/parents’ history, as well as due to the comparison of samples with different socioeconomic and cultural backgrounds using different diagnostic tools [4].

This study aims to estimate the prevalence of SB in a sample of pediatric patients of a Portuguese dentistry service through parent/caregiver reporting and to assess its increase or decrease with age.

## 2. Materials and Methods

An observational retrospective study was conducted at the Lisbon Pediatric Dentistry Service (Serviço Odontopediátrico de Lisboa—SOL) according to the guidelines of the Declaration of Helsinki and approved by the Ethics Committee for Health of CMRA (Centro de Medicina de Reabilitação de Alcoitão)/SCML (Santa Casa da Misericórdia de Lisboa) (protocol code SOL2021_001 approved on 14 September 2021). This pediatric dentistry service has 12,000 pediatric patients and more than 40,000 appointments per year. It is a community service that has been providing pediatric dental care to the municipal population of Lisbon since August 2019. This service is guaranteed by an organization (Santa Casa da Misericórdia de Lisboa) supervised by the Portuguese Ministry of Solidarity and Social Security, complementary to the Portuguese National Health Service. 

In the first appointment, informed consent is required from the parents/caregivers, and a very detailed and standardized questionnaire (anamnesis) is performed and recorded by dentists in the clinical software NoviGest (Tactis, Porto, Portugal) ^®^. In order to collect parents/caregivers’ perception of SB, a single question is asked: presence/absence of SB via the existence of dental tightening or audible noises related to teeth grinding. All parents/caregivers’ doubts and questions are clarified. Parents/caregivers’ answers are recorded as a dichotomous variable (yes/no).

Five assessors analyzed the clinical records of all patients using Novigest^®^ software. Children between 0 and 17 years old with their first appointment between August 2019 and December 2020 were included in the analysis. Children double-registered in the Novigest^®^ software with teleconsultation or emergency appointment were excluded. The information related to SB (yes/no) was extracted to an Excel sheet, excluding any personal data that could further lead to the identification of patients.

In summary, the inclusion criteria were: age < 18; first appointment between August 2019 and December 2020 and presence of SB via the existence of dental tightening or audible noises related to teeth grinding perceived by patients’ parents/caregivers. Exclusion criteria: children double-registered in the Novigest^®^ with teleconsultations and emergency appointments. Only the first appointment of the children was considered (single-visit retrospective observational study). No clinical examination or polysomnography was performed to confirm parents/caregivers’ perceptions of SB.

According to the purpose of the study, the sample size was calculated following the approach described in Daniel WW, 1999 [17]. Thus, assuming a prevalence of SB of 49.6% (the worst scenario in Machado et al., 2014 [4]), a 5% acceptable error rate, and 95% confidence interval (CI), the minimum number of patients to include in this study should be 385. After excluding the patients double-registered in the Novigest^®^ software with teleconsultations or emergency appointments, a total of 1900 patients was obtained.

Descriptive analysis was performed. Prevalence of SB was calculated at a global level and for the age and sex sub-groups with a 95% confidence interval. Statistical analyses were performed using STATA version 16.1 (Stata Corp LP, College Station, TX, USA).

## 3. Results

A total of 1900 patients were included, 961 (50.6%) and 939 (49.4%) of whom were male and female, respectively (Table 1). The majority, comprising 822 (43.3%) patients, were between 12 and 17 years old, followed by 725 (38.2%) between 7 and 11 years old and 353 (18.6%) between 0 and 6 years old. The age distribution was very similar when analyzed by sex.

Among the included patients, 334 (17.6%, 95%CI 15.9–19.4%) were identified as having SB (Table 2), corresponding to 182 (18.9%, 95% CI 16.5–21.6%) males and 152 (16.2%, 95% CI 13.9–18.7%) females. SB was identified in 73 (20.7%, 95% CI 16.6–25.3%) patients between 0 and 6 years old, 141 (19.4%, 95% CI 16.6–22.5%) patients between 7 and 11 years old, and 120 (14.6%, 95% CI 12.3–17.2%) between 12 and 17 years old (Table 2).

Considering the 939 female patients included, SB was identified in 34 (19.2%, 95% CI 13.7–25.8%) patients between 0 and 6 years old, 60 (16.9%, 95% CI 13.1–21.2%) female patients between 7 and 11 years old, and 58 (14.3%, 95% CI 11.0–18.1%) between 12 and 17 years old. Of the total of 961 male patients, SB was identified in 39 (22.2%, 95% CI 16.3–29.0%) between 0 and 6 years old, 81 (22.0%, 95% CI 17.8–26.5%) between 7 and 11 years old, and 62 (14.9%, 95% CI 11.6–18.7%) between 12 and 17 years old. There is no statistical evidence to conclude that the prevalence is different between females and males within the different considered age groups at a significance level of α = 0.05.

## 4. Discussion

This study aimed to estimate the prevalence of SB in a sample of pediatric patients of a Portuguese pediatric dentistry service and to assess whether it increases with age. The prevalence of SB was estimated based on patients’ parents/caregivers’ reports only, with no diagnosis by professionals, and no clinical examination or polysomnography was performed to confirm parents/caregivers’ perceptions of SB. We found a prevalence of SB of 17.6%, which was slightly higher among male patients. We also observed a decrease in SB with age.

Information provided by parents/caregivers (legal tutors) during the first consultation concerning the presence or absence of SB was included in this study. This was registered regarding parents/caregivers’ reporting method for SB: the existence of dental tightening or audible noises related to teeth grinding. It was observed that a multiple observation report had a better correlation with polysomnographic findings [18].

In our study, 17.6% of pediatric patients presented a diagnosis of SB, and we are 95% confident that the global prevalence of SB in pediatric patients is between 15.9% and 19.4%. A systematic review of the literature reported a prevalence ranging from 5.9% to 49.6% depending on the diagnostic criteria, which remains the main difficulty when establishing comparisons between different authors and contexts [4]. It is also important to consider the impact of the geographical variation of parental-reported SB. In developed countries, where children have their own bedroom or have less tension, among other cultural factors, the prevalence of SB was lower than in developing countries [19].

In our sample, SB was identified in 20.7% of patients between 0 and 6 years old and in 19.4% of patients between 7 and 11 years old. In patients between 12 and 17 years old, SB was identified in 14.3%, showing a decrease in SB with age. These results are in accordance with the trend reported by other authors, although at a higher prevalence [4,20]. A systematic review performed by Machado et al. [4] identified a prevalence of SB between 5.9% and 36.8% in pre-school children and 49.6% in first graders. Manfredini et al. [20] reported a prevalence ranging from 3.5% to 8.5% in children up to 5 years old and 34.7% to 40% in children between 7 and 10 years old. The literature suggests a decrease in SB with age, supporting the belief that SB behavior progressively decreases by the age of 9–10 years old and that the majority of SB children do not continue bruxing through adolescence and adulthood: Saulue et al. [21] registered a prevalence of 15.0% in patients younger than 12 years and 12.4% in those older than 13 years [4,15,20]. Other authors suggested a prevalence of 14–20% during childhood, which was reduced to 8% in adulthood and 3% in older adults [15].

We found that 18.9% of males and 16.2% of females presented with SB, illustrating a slightly higher prevalence of SB among male pediatric patients regardless of age. Nonetheless, this difference was not statistically significant (*p* = 0.121). The systematic review by Manfredini et al. [20] reported no sex differences, while other authors reported that SB affects more male patients than female patients [4].

These data need some considerations in their interpretation. Standardized diagnostic criteria were not considered when assessing the prevalence of SB. Thus, further studies addressing the patients’ awareness, clinical examination, practical issues related to the differences observed between age groups, and the parents’ educational level should be conducted [2,21,22]. It is also important to take into account the subjective nature of patients’ parent/caregiver reports, since SB report is based on the awareness of its signs [21], and no clinical examination was performed to confirm the findings. In addition, the majority of pediatric patients sleep in a different room than their parents/caregivers, who may, therefore, not be aware of SB signs [2], with no detection of clenching that is silent, and a better reporting rate was associated with opened doors [22]. Older children, including teenagers, probably do not share the same room as their parents, and practical issues, including whether the doors are opened or not are not known, which can affect the reporting of the episodes for patients of older ages and, consequently, be an important limitation of our study. Moreover, the target population of patients assessed is generically characterized by low-income families, which may also compound a selection bias. Although the role of the income status is not relevant in bruxism proxy reporting [23], other situations associated with a low-income status should not be discarded as potential risk factors for bruxism in this sample of patients. Finally, the size of our sample is comparable to other studies [20]. However, we included patients with a large age range. Therefore, it is of utmost importance to perform a study with a larger dimension of patients in each age group, adjust the age range groups, and complement the parents/caregivers’ report with clinical examination and polysomnography, which is the gold standard in SB diagnosis.

## 5. Conclusions

Concerning the pediatric patients that participated in the study, 17.6% (95% CI 15.9–19.4%) presented SB. The prevalence was slightly higher among male patients and decreased with age. However, further studies are needed to overcome the limitations of this study and establish correlations to eliminate confounding factors, such as the presence of comorbidities, bedroom sharing between parents and children, and the routine of opening or closing bedroom doors. As a final remark, asking parents about SB is a topic that should be addressed in pediatric dentistry appointments in order to understand and improve parental awareness about SB.

## Figures and Tables

**Table 1 ijerph-19-07823-t001:** Demographic characteristics of patients with sleep bruxism (SB). The total number of subjects analyzed was 1900. n, number of subjects in the category.

Characteristic	Total *n* (%)
Sex	
Female	939 (49.4)
Male	961 (50.6)
Age Group (years)	
0–6	353 (18.6)
7–11	725 (38.2)
12–17	822 (43.3)

**Table 2 ijerph-19-07823-t002:** Prevalence of sleep bruxism (SB) with 95% confidence interval (CI): overall, by sex, and by age. N, total number of subjects in the category; n, number of subjects with SB in the category.

	Female n/N (%) (95% CI)	Male n/N (%) (95% CI)	Total n/N (%) (95% CI)	*p*-Value ^1^ (Female vs. Male)
Overall prevalence	152/939 (16.2) (3.9; 18.7)	182/961 (18.9) (16.5; 21.6)	334/1900 (17.6) (15.9; 19.4)	0.121
Prevalence 0–6 years	34/177 (19.2) (13.7; 25.8)	39/176 (22.2) (16.3; 29.0)	73/353 (20.7) 16.6; 25.3)	0.484
Prevalence 7–11 years	60/356 (16.9) (13.1; 21.2)	81/369 (22.0) (17.8; 26.5)	141/725 (19.4) (16.6; 22.5)	0.084
Prevalence 12–17 years	58/406 (14.3) (11.0; 18.1)	62/416 (14.9) (11.6; 18.7)	120/822 (14.6) (12.3; 17.2)	0.810

^1^ Calculated using the two-sided Z-Test.

## Data Availability

Data regarding this study can be shared upon request to the corresponding author.

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
