# Peer review of "Prevalence of Sleep Bruxism Reported by Parents/Caregivers in a Portuguese Pediatric Dentistry Service: A Retrospective Study"

_ijerph, 2022, doi:10.3390/ijerph19137823_

Round 1
Reviewer 1 Report
Bruxism is a dental problem that affects both adults and children. It is an important issue, the exploration of which may in the future increase social awareness and improve dental care, especially among the youngest. So the article touches on very important issues.
The number of analyzed patients is impressive, but due to the fact that the study covers only information provided by parents and dentists, it is difficult to call the text an article - rather, it is the result of a survey subject to basic statistical analysis.
The lack of any documentation in the form of photos reduces the quality of the article even more.
If the editor decides to publish, take a long time to properly format the tables and text.
However, I believe that the article - although it contains very interesting information, should not be published in a journal of such high rank due to its lack of scientific character.
Author Response
Point 1: The lack of any documentation in the form of photos reduces the quality of the article even more.
Response 1: The aim of our work was to estimate the prevalence of sleep bruxism (SB) based only on parents/caregivers’ reports. Thus, no photos were taken to the bruxism signs, since it was not the purpose of the study. With this sample, we believe that these results can provide interesting data on bruxism prevalence, although we must overcome confounding factors (described in our article) to obtain more accurate answers on the matter of SB.
Point 2: If the editor decides to publish, take a long time to properly format the tables and text.
Response 2: Tables and text were formatted according to MDPI guidelines.
Reviewer 2 Report
The authors aimed to conduct a retrospective observational study based on information provided by patients’ parents/caregivers, between August 2019 and December 2020 about the prevalence of sleep bruxism (SB), and assess its trend with age in a sample of pediatric patients who frequented Lisbon Pediatric Dentistry Service (SOL).
The study covers some issues that have been overlooked in other similar topics. The structure of the manuscript appears adequate and well divided in the sections. Moreover, the study is easy to follow, but some issues should be improved. Some of the comments that would improve the overall quality of the study are:
- Authors must pay attention to the technical terms acronyms they used in the text.
- English language needs to be revised.
- Conclusion Section: This paragraph needs to be improved. Please also add some "take-home message".
Author Response
Point 1: Authors must pay attention to the technical terms acronyms they used in the text.
Response 1: We have revised the technical terms acronyms, mainly in Methods section. We have followed MDPI instructions for abbreviations: the Acronyms/Abbreviations/Initialisms should be defined the first time they appear in each of three sections: the abstract; the main text; the first figure or table. When defined for the first time, the acronym/abbreviation/initialism should be added in parentheses after the written-out form.
Point 2: English language needs to be revised.
Response 2: An english review was carried out.
Point 3: Conclusion Section: This paragraph needs to be improved. Please also add some "take-home message".
Response 3: A take-home message was added to the Conclusion section (lines 215-217).
Reviewer 3 Report
Dear Authors,
The article: 'Prevalence of sleep bruxism in a Portuguese pediatric dentistry service: a report by parents/caregivers' was to determine SB prevalence and assess its trend with age in a sample of pediatric patients who frequented Lisbon Pediatric Dentistry Service (SOL).
English language and style must be corrected.
Numerous punctuation mistakes should be corrected.
Merge affiliations.
The introduction is well written. The purpose of the work should be clearly defined.
Use the same text style according to the MDPI guidelines.
Materials and methods
Add subsections. Add inclusion and exclusion criteria.
Separately add statistics.
Institutional Review Board Statement add in materias and methods.
Tables should be prepared using MDPI guidelines.
The discussion is well written
page 5: ' Carra et al. registered a prevalence of 15.0% in patients younger
than 12 years and 12.4% in those older than 13 years [4,15,20].' should be ' Saulue et al. [22] registered a prevalence of 15.0% in patients younger than 12 years and 12.4% in those older than 13 years [4,15,20].' Check citation in the text!!!
Discussion is clearly presented.
Add table with abbeviations befeore references.
Add contribution of authors at the end of the text.
References should be prepared in accordance with the MDPI guidelines.
Article can be accepted after major revision.
Author Response
Point 1: English language and style must be corrected. Numerous punctuation mistakes should be corrected.
Response 1: Style was corrected.
Point 2: Merge affiliations.
Response 2: Done.
Point 3: The introduction is well written. The purpose of the work should be clearly defined.
Response 3: The purpose of the work is defined in lines 79-81.
Point 4: Use the same text style according to the MDPI guidelines.
Response 4: Style was corrected.
Point 4: Materials and methods. Add subsections. Add inclusion and exclusion criteria.
Response 4: We revised the methods section in order to address clearly the inclusion and exclusion criteria. The inclusion criteria were: age < 18; first appointment between August 2019 and December 2020; presence of SB, by the existence of dental tightening or audible noises related to teeth grinding precepted by patients’ parents/caregivers. Exclusion criteria: Children double registered in the Novigest®, with teleconsultations and emergency appointments. According to MDPI guidelines the Materials and methods section is not supposed to have subsections.
Point 5: Separately add statistics.
Response 5: According to MDPI guidelines the Materials and methods section is not supposed to have subsections.
Point 6: Institutional Review Board Statement add in materials and methods.
Response 6: Added as suggested. (line 83-86)
Point 7: Tables should be prepared using MDPI guidelines.
Response 7: Tables were formatted according to MDPI guidelines.
Ponint 8: page 5: ' Carra et al. registered a prevalence of 15.0% in patients younger than 12 years and 12.4% in those older than 13 years [4,15,20].' should be ' Saulue et al. [22] registered a prevalence of 15.0% in patients younger than 12 years and 12.4% in those older than 13 years [4,15,20].' Check citation in the text!!!
Response 8: Corrected
Point 9: Add table with abbeviations befeore references.
Response 9: We have followed MDPI instructions for abbreviations: the Acronyms/Abbreviations/Initialisms should be defined the first time they appear in each of three sections: the abstract; the main text; the first figure or table. When defined for the first time, the acronym/abbreviation/initialism should be added in parentheses after the written-out form. Thus, no table was added.
Point 10: Add contribution of authors at the end of the text.
Response 10: Authors contributions are stated after Conclusion section, according to MDPI guidelines. (lines 219-224)
Point 11: References should be prepared in accordance with the MDPI guidelines.
Response 11: References were updated according MDPI guidelines, using the American Chemical Society style.
Reviewer 4 Report
Manuscript of interest with considerable sample size.
Before being published it needs a major revision.
Abstract: it must be specified that the study was carried out during the Covid-19 pandemic, it should be specified in the title of the manuscript itself.
Keywords: few, add more specifics.
Introduction: Add minimally invasive approach during the pandemic (as described in the review by Prof Scribante et al. Published in the same journal as the submission) DOI
10.3390/jcm9123914
How were the patients monitored?
Materials and methods: the inclusion and exclusion criteria used by patients and the frequency of visits should be extended.
Results: well described
Discussion add the active proncypes used in home oral hygiene products, if they were patients destined for orthodontic care and a PROGRESSIVE fluoride suspension as described by the research group of Prof Farronato, I am attaching the reference: DOI
10.4103/2278-0203.197392
Conclusion add preventive effects
Bibliography: add suggested references
Author Response
Point 1: Abstract: it must be specified that the study was carried out during the Covid-19 pandemic, it should be specified in the title of the manuscript itself.
Response 1: The study was written during Covid-19 pandemic, but the appointments in which we collected the results took place before. The patients were not monitored since this was a single visit retrospective clinical study.
Point 2: Keywords: few, add more specifics.
Response 2: Added prevalence.
Point 3: Introduction: Add minimally invasive approach during the pandemic (as described in the review by Prof Scribante et al. Published in the same journal as the submission) DOI 10.3390/jcm9123914
Response 3: Do not seem related to the present paper, as a retrospective observational study about prevalence of bruxism in children.
Point 4: How were the patients monitored?
Response 4: Patients were not monitored, since this was a single visit retrospective clinical study.
Point 5: Materials and methods: the inclusion and exclusion criteria used by patients and the frequency of visits should be extended.
Response 5: We revised the methods section in order to address clearly the inclusion and exclusion criteria. The inclusion criteria were: age < 18; first appointment between August 2019 and December 2020; presence of SB, by the existence of dental tightening or audible noises related to teeth grinding precepted by patients’ parents/caregivers. Exclusion criteria: Children double registered in the Novigest®, with teleconsultations and emergency appointments. Only the first appointment of the children was considered (single visit)
Point 6: Discussion add the active proncypes used in home oral hygiene products, if they were patients destined for orthodontic care and a PROGRESSIVE fluoride suspension as described by the research group of Prof Farronato, I am attaching the reference: DOI 10.4103/2278-0203.197392.
Response 6: The use of oral hygiene products do not seem related to the present paper, as a retrospective observational study about prevalence of bruxism in children. The reference suggested is not in scope of our article.
Point 7: Conclusion add preventive effects
Response 7: Do not seem related to the present paper, as a retrospective observational study about prevalence of bruxism in children.
Point 8: Bibliography: add suggested references
Response 8: Overall, the comments provided by this reviewer don’t seem to be for our article. As a retrospective observational study about prevalence of bruxism in children, the references suggested are not in scope of our article.
Reviewer 5 Report
I would like to congratulation the authors for this study and suggest some minor adjustments.
In the abstract section on line 26, I suggest including “and without clinical examination or polysomnography examination” after “parents/caregivers”.
The authors included that no clinical examination or polysomnography was performed in the Materials and Methods section, but it is also important to emphasize this information in the Discussion section and explain the limitations of the study.
If the authors are able to perform a clinical examination/polysomnography for at least part of these patients, they will be able to obtain more information to compare and discuss both results. It could also be an idea for a future study.
Author Response
Point 1: In the abstract section on line 26, I suggest including “and without clinical examination or polysomnography examination” after “parents/caregivers”.
Response 1: Added as suggested. (line 16)
Point 2: The authors included that no clinical examination or polysomnography was performed in the Materials and Methods section, but it is also important to emphasize this information in the Discussion section and explain the limitations of the study.
Response 2: Discussion section reformulated to include that no clinical examination or polysomnography was performed. The limitations of the study are presented in lines 205-207.
Point 3: If the authors are able to perform a clinical examination/polysomnography for at least part of these patients, they will be able to obtain more information to compare and discuss both results. It could also be an idea for a future study.
Response 3: This is a great suggestion for a future study and we will take it in consideration.
Round 2
Reviewer 1 Report
Accept in present form
Author Response
Response to Reviewer 1 Comments
Point 1: Accept in present form
Response 1: We would like to thank you for your contributions and acceptance after we have revised the paper according to your comments.
Reviewer 3 Report
article can be accepted after Editor decision
Author Response
Response to Reviewer 3 Comments
Point 1: article can be accepted after Editor decision
Response 1: We would like to thank you for your contributions and acceptance after we have revised the paper according to your comments.
Reviewer 4 Report
The authors did not make major changes to the manuscript, inappropriately pointing to the comments suggested, even if it is an observational study the considerations were appropriate for the context in which the study was inserted, even if it is about bruxism there are always of the conditions of oral hygiene to be maintained and the clinician is required to know.
Author Response
Response to Reviewer 4 Comments:
Point1: "The authors did not make major changes to the manuscript, inappropriately pointing to the comments suggested, even if it is an observational study the considerations were appropriate for the context in which the study was inserted, even if it is about bruxism there are always of the conditions of oral hygiene to be maintained and the clinician is required to know."
Response 1: Dear reviewer, we have the most respect for the area of ​​oral hygiene and are fully aware of the importance of preventive care. That's why in our vast clinical and investigation team we include oral hygienists. We think your comment on this matter is unfair.
As for the article, we must clarify and we will divide the clarification into two points:
At first we consider it strange and unfair that from the first review to the second you have changed some points from "can be improved" to "must be improved", without any changes having been made other than our comments on your review.It's strange, unjust and unjustifiable and jeopardizes the work of several months of a vast team that is always giving its best.
Secondly, the question of the articles that you suggested and considered relevant to mention. In this case and after the entire team reviewed and discussed the articles mentioned “Timing considerations on the shear bond strength of orthodontic brackets after topical fluoride varnish applications” DOI 10.4103/2278-0203.197392" and "Bio-Inspired Systems in Nonsurgical Periodontal Therapy to Reduce Contaminated Aerosol during COVID-19: A Comprehensive and Bibliometric Review” DOI 10.3390/JCM9123914.", we still do not understand the need and usefulness of its inclusion, since it has nothing to do with the subject under study in this article, regardless of the value of the same that is not in question.
In the last review report, it does not raise other questions or ask for concrete changes, other than those raised in the first review and which relate to the reference to the two articles and to the issue of the pandemic period that we have already justified do not make sense. For this reason, we have nothing more to add than to appeal to your common sense and respect for the months-long work of a vast team of clinicians.
For these reasons, we inform you that we are not going to change the article with these suggestions, because, as we have already mentioned, they are not related or make sense to include in our investigation. Perhaps in a future article related to oral hygiene matters (which we also studied) we could include them.
This manuscript is a resubmission of an earlier submission. The following is a list of the peer review reports and author responses from that submission.
Round 1
Reviewer 1 Report
This study used a retrospective observational data to investigate sleep bruxism (SB) prevalence and evaluated its trend with age in a group of pediatric patients who frequently visited a pediatric dentistry service. The authors found that the SB prevalence was slightly higher in male than in female patients (18.9% vs 16.2%). The authors also concluded that standardized diagnostic criteria for SB is needed and the subjectivity of the data collection methods needs to be improved.
The research data were obtained from the pediatric dental records which were reported by patients’ parents or caregivers during the first dental clinical visit. No standardized clinical criteria were made in the data collection. Information bias is a major concern. In addition, the data was collected by only one pediatric dentistry service. We are not aware whither the patients were continuously or intermittently collected? Selection bias is also a major issue. Overall, this study is not novel and has some flaws.
Reviewer 2 Report
This study was based on parental reports of sleep Bruxism in their children regardless of their age (young or adolescents). Clinical examination to confirm the presence of any signs that can raise the suspicion of the presence or confirm the habit was not considered. It is also unclear from the study if the parents received an explanation about the condition. The findings from such a study are highly debatable and the authors acknowledge that in the discussion. The method of data collection is not addressed nor the questions which addressed the presence/absence of bruxism are clearly described. The authors state that 5 calibrated examiners collected the information (line 105) although it is unclear the calibration was for what as no clinical examination of the children was performed. Also, the inclusion criteria for the children are not addressed.
The method of sample size calculation was not given and descriptive reporting of the results was only considered which should be insufficient to draw any conclusion particularly for comparisons regarding age group and gender.